# CLOSING THE GAP BETWEEN TD LEARNING AND SUPERVISED LEARNING – A GENERALISATION POINT OF VIEW

**Raj Ghugare**[1]     **Matthieu Geist**[2]     **Glen Berseth**[1*]     **Benjamin Eysenbach**[3*]

[1]Mila, Université de Montréal     [2]Google DeepMind     [3]Princeton University

raj.ghugare@mila.quebec

## ABSTRACT

Some reinforcement learning (RL) algorithms can *stitch* pieces of experience to solve a task never seen before during training. This oft-sought property is one of the few ways in which RL methods based on dynamic-programming differ from RL methods based on supervised-learning (SL). Yet, certain RL methods based on off-the-shelf SL algorithms achieve excellent results without an explicit mechanism for stitching; it remains unclear whether those methods forgo this important stitching property. This paper studies this question for the problems of achieving a target goal state and achieving a target return value. Our main result is to show that the stitching property corresponds to a form of combinatorial generalization: after training on a distribution of (state, goal) pairs, one would like to evaluate on (state, goal) pairs not seen together in the training data. Our analysis shows that this sort of generalization is different from *i.i.d.* generalization. This connection between stitching and generalisation reveals why we should not expect SL-based RL methods to perform stitching, even in the limit of large datasets and models. Based on this analysis, we construct new datasets to explicitly test for this property, revealing that SL-based methods lack this stitching property and hence fail to perform combinatorial generalization. Nonetheless, the connection between stitching and combinatorial generalisation also suggests a simple remedy for improving generalisation in SL: data augmentation. We propose a *temporal* data augmentation and demonstrate that adding it to SL-based methods enables them to successfully complete tasks not seen together during training. On a high level, this connection illustrates the importance of combinatorial generalization for data efficiency in time-series data beyond tasks beyond RL, like audio, video, or text.

## 1 INTRODUCTION

Many recent methods view RL as a purely SL problem of mapping input states and desired goals, to optimal actions [1–3]. These methods have gained a lot of attention due to their simplicity and scalability [4]. These methods sample a goal $g$ (or a return $r$) from the dataset, which was previously encountered after taking an action $a$ from a state $s$, and then imitate $a$ by treating it as an optimal label for reaching $g$ (or achieving return $r$) from $s$. These methods, collectively known as outcome conditional behavioral cloning algorithms (OCBC), achieve excellent results on common benchmarks [3]. However, at a fundamental level, there are some important differences between RL and SL. This paper studies one of those differences: the capability of some RL algorithms to stitch together pieces of experience to solve a task never seen during training. While some papers have claimed that some OCBC approaches already have this stitching property [2], both our theoretical and empirical analyses suggest some important limitations of these prior claims.

The stitching property [5] is common among RL algorithms that perform dynamic programming (e.g., DQN [6], DDPG [7], TD3 [8], IQL [9]). It is often credited for multiple properties of dynamic programming algorithms like superior data efficiency and off policy reasoning (See Section 4 for detailed discussion). Importantly, we show that stitching also allows for a third property – the ability to infer solutions to a combinatorial number of tasks during test time, like navigating between certain state-goal pairs that never appear together (but do appear separately) during training. An example of stitching is that humans don't need access to optimal actions to go from an airport to new tourist places; they can use their previous knowledge to navigate to a taxi-stand, which would take them to any

---

*Equal advising.

location. But, purely supervised approaches to sequential problems like RL, do not explicitly take such temporal relationships into account. Even in other sequential domains like language, a large body of work is dedicated to study the combinatorial generalisation abilities of large language models [10–13]. Our work shows that combinatorial generalisation is also required to solve tasks in the context of RL.

We start by formalising stitching as a form of combinatorial generalisation. We observe that when data are collected from a mixture of policies, there can be certain (state, goal) pairs that are never visited in the same trajectory, despite being frequented in separate trajectories. Information from multiple trajectories should be *stitched* to complete these tasks. Because such tasks (state-goal pairs) are seen separately, but never together, we call the ability of algorithms to perform these tasks as combinatorial generalisation. This connection further motivates an inspiration from SL; if generalisation is the problem, then data augmentation is likely an effective approach [14]. We propose a form of temporal data augmentation for OCBC methods so that they acquire this stitching property and succeed in navigating between unseen (start, goal) pairs or achieving greater returns than the offline dataset. This form of data augmentation involves *time*, rather than random cropping or shifting colors. Intuitively, temporal data augmentation augments the original goal, with a new goal sampled from a different overlapping trajectory in the offline dataset. This data augmentation scheme does require an estimate of distance between states to detect overlapping trajectories. We demonstrate that this data augmentation is theoretically backed, and empirically endows OCBC algorithms with the stitching property on difficult state-based and image-based tasks.

Our primary contribution is to provide a formal framework for studying stitching as a form of combinatorial generalisation. Because of this connection, we hypothesize that OCBC methods do not perform stitching. Perhaps surprisingly, simply increasing the volume of data does not guarantee this sort of combinatorial generalization. Our empirical results support the theory: we demonstrate that prior RL methods based on SL (DT [2] and RvS [3]) fail to perform stitching, even when trained on abundant quantities of data. Our experiments reveal a subtle consideration with the common D4RL datasets [15]: while these datasets are purported to test for exactly this sort of combinatorial generalization, data analysis reveals that "unseen" (state, goal) pairs do actually appear in the dataset. Thus, our experiments are run on a new variant of these datasets that we constructed for this paper to explicitly test for combinatorial generalization [1]. On 10 different environments, including both state and image based tasks, and goal and return conditioning, adding data augmentation improves the generalisation capabilities of SL approaches by up to a factor of 2.5.

## 2 RELATED WORK

**Prior methods that do some form of explicit stitching.** Previous work on stitching abilities of SL algorithms have conflicting claims. The DT paper [2] shows experiments where their SL-based method performs stitching. On the contrary, [16] provide an example where SL algorithms do not perform stitching. RvS [3] shows that a simple SL-based algorithm can surpass the performance of TD algorithms. In tabular settings, [17] show that the benefits of TD-learning arise from trajectory stitching. We provide a formal definition of stitching as a form of combinatorial generalisation. In contrast, generalisation in RL has been generally associated with making correct predictions for unseen but similar states and actions [18–20], planning [21], ignoring irrelevant details [22–24], or robustness towards changes in the reward or transition dynamics [25–27].

**Offline RL datasets.** A large amount of work is done to build offline RL datasets. [15] provided a first standard offline RL benchmark, [28] provide exploratory offline datasets to underscore the importance of diverse data, [29, 30] focus on data efficiency and real world deployement and [31] provide benchmarks that also compare the online evaluation budget of offline RL algorithms. Although many offline RL papers informally allude to stitching, we devise new offline RL datasets that precisely test the stitching abilities of offline RL algorithms.

**Data augmentation in RL.** Data augmentation has been proposed as a remedy to improve generalisation in RL [32–38], akin to SL [39]. Perhaps the most similar prior work are the ones which use dynamic programming to augment existing trajectories to improve the performance properties of SL algorithms [40–42]. However, because these methods still require dynamic programming, they don't have the same simplicity that make SL algorithms appealing in the first place.

---

[1]Open sourced code and data is available: `https://github.com/RajGhugare19/stitching-is-combinatorial-generalisation`

## 3 PRELIMINARIES

**Controlled Markov processes.** We will study the problem of goal-conditioned RL in a controlled Markov process with states $s \in \mathcal{S}$ and actions $a \in \mathcal{A}$. The dynamics are $p(s' \mid s, a)$, the initial state distribution is $p_0(s_0)$, the discount factor is $\gamma$. The policy $\pi(a, \mid s, g)$ is conditioned on a pair of state and goal $s, g \in \mathcal{S}$. For a policy $\pi$, define $p_t^\pi(s_t \mid s_0)$ as the distribution over states visited after exactly $t$ steps. We can then define the discounted state occupancy distribution and its conditional counterpart as

$$p_+^\pi(s_{t+} = g) \triangleq \mathbb{E}_{s \sim p_0(s_0)} \left[ p_+^\pi(s_{t+} = g \mid s_0 = s) \right], \tag{1}$$

$$p_+^\pi(s_{t+} = g \mid s_0 = s) \triangleq (1 - \gamma) \sum_{t=0}^\infty \gamma^t p_t^\pi(s_t = g \mid s_0 = s), \tag{2}$$

where $s_{t+}$ is the variable that specifies a future state corresponding to the discounted state occupancy distribution. Given a state-goal pair $s, g \sim p_{\text{test}}(s, g)$ at test time, the task of the policy is to maximise the probability of reaching the goal $g$ in the future

$$\max_\pi J(\pi), \quad \text{where} \quad J(\pi) = \mathbb{E}_{s,g \sim p_{\text{test}}(s,g)} \left[ p_+^\pi(s_{t+} = g \mid s_0 = s) \right]. \tag{3}$$

**Data collection.** Our work focuses on the offline RL setting where the agent has access to a fixed dataset of $N$ trajectories $\mathcal{D} = (\{s_0^i, a_0^i, ..\})_{i=1}^N$. Our theoretical analysis will assume that the dataset is collected by a set of policies $\{\beta(a \mid s, h)\}$, where $h$ specifies some context. For example, $h$ could reflect different goals, different language instructions, different users or even different start state distributions. Precisely, we assume that the data was collected by first sampling a context from a distribution $p(h)$, and then sampling a trajectory from the corresponding policy $\beta(a \mid s, h)$. We will use the shorthand notation $\beta_h(\cdot \mid \cdot) = \beta(\cdot \mid \cdot, h)$ to denote the data collecting policy conditioned on context $h$. Trajectories are assumed to be stored without $h$, hence the context denotes all hidden information that the true data collection policies used to collect the data.

This setup of collecting data corresponds to a mixture of Markovian policies[2]. There is a classic result saying that, for every such *mixture* of Markovian policies, there exists a Markovian policy that has the same discounted state occupancy measure.

**Lemma 3.1** (Rephrased from Theorem 2.8 of [43], Theorem 6.1 of [44])**.** *Let a set of context-conditioned policies $\{\beta_h(a \mid s)\}$ and distribution over contexts $p(h)$ be given. There exists a Markovian policy $\beta(a \mid s)$ such that it has the same discounted state occupancy measure as the mixture of policies:*

$$p_+^\beta(s_{t+}) = \mathbb{E}_{p(h)} \left[ p_+^{\beta_h}(s_{t+}) \right]. \tag{4}$$

The policy $\beta(a \mid s)$ is simple to construct mathematically as follows. For data collected from the mixture of context conditioned policies, let $p^\beta(h \mid s)$ be the distribution over the context given that the policy arrived in state $s$.

$$\beta(a \mid s) \triangleq \sum_h \beta_h(a \mid s) p^\beta(h \mid s). \tag{5}$$

Theorem 6.1 [44] proves the correctness of this construction. The policy $\beta(a \mid s)$ is also easy to construct empirically – simply perform behavioral cloning (BC) on data aggregated from the set of policies. We will hence call this policy the BC policy.

**Outcome Conditional behavioral cloning (OCBC).** While our theoretical analysis will consider generalisation abstracted away from any particular RL algorithm, we will present empirical results using a simple and popular class of goal-conditioned RL methods: Outcome conditional behavioral cloning [45] (DT [2], URL [1], RvS [3], GCSL [46] and many others [47, 48]). These methods take as input a dataset of trajectories $\mathcal{D} = (\{s_0^i, a_0^i, ..\})_{i=1}^N$ and learn a goal-conditioned policy $\pi(a \mid s, g)$ using a maximum likelihood objective:

$$\max_{\pi(\cdot \mid \cdot, \cdot)} \mathbb{E}_{(s,a,g) \sim \mathcal{D}} \left[ \log \pi(a \mid s, g) \right]. \tag{6}$$

---

[2]Note that the mixture is at the level of trajectories, not at the level of individual actions.

The sampling above can be done by first sampling a trajectory from the dataset (uniformly at random), then sampling a (state, action) pair from that trajectory, and setting the goal to be a random state that occurred later in that same trajectory. If we incorporate our data collecting assumptions, then this sampling can be written as

$$\max_{\pi(\cdot|\cdot,\cdot)} \mathbb{E}_{h\sim p(h)} \Big[ \mathbb{E}_{\substack{a,s\sim\beta_h(a|s),\, p_+^{\beta_h}(s) \\ s_{t+}\sim p_+^{\beta_h}(s_{t+}|s,a)}} \big[ \log \pi(a \mid s, s_{t+}) \big] \Big]. \tag{7}$$

## 4 "Stitching" as a form of combinatorial generalisation

Before concretely defining stitching, we will describe three desirable properties that are colloquially associated with "stitching" and the learning dynamics of TD methods. *(Property 1)* The ability to select infrequently seen paths that are more optimal than frequent ones. While a shorter trajectory between the state and the goal may occur infrequently in the dataset, TD methods can find more examples of this trajectory by recombining pieces of different trajectories, thanks to dynamic programming. This property is enjoyed by both SARSA (expectation over actions) and Q-learning (max over actions) methods, and is primarily associated with the sample efficiency of learning. *(Property 2)* The ability to evaluate policies different from those which collected the data, and perform multiple steps of policy improvement. This property is unique to Q-learning. *(Property 3)* Temporal difference methods (both Q-learning and SARSA) can also recombine trajectories to find paths between states never seen together during training. This property is different from the first property in that it is not a matter of data efficiency – temporal difference methods can find paths that will never be sampled from the data collecting policies, even if given infinite samples. All three of these properties are colloquially referred to as "stitching" in the literature. While these properties are not entirely orthogonal, they are distinct: certain algorithms may have just some of these properties. Obtaining all these properties in a simpler (than TD) framework is difficult, and it remains unclear whether OCBC methods possess any of them. To better understand the differences and similarities this study focuses on the third property. We formalize this property as a form of generalisation, which we will refer to as combinatorial generalisation.

Defining combinatorial generalisation will allow us to analyze if and when OCBC methods perform stitching, both theoretically (this section) and experimentally (Section 6). Intuitively, *combinatorial generalisation* looks at connecting states and goals, which are never seen together in the same trajectory, but where a path between them is possible using the information present in different trajectories. It therefore tests a form of "stitching" [15, 49], akin to "combinatoral generalisation" [50–52]. To define this generalisation, we will specify a training distribution and testing distribution. The training distribution corresponds to sampling a context $h \sim p(h)$ and then sampling an $(s, g)$ pair from the corresponding policy $\beta_h$. This is exactly how OCBC methods are trained in practice (Section 3). The testing distribution corresponds to sampling an $(s, g)$ pair from the BC policy $\beta(a \mid s)$ defined in Equation (5). For each distribution, we will measure the performance $f^{\pi(\cdot|\cdot,g)}(s, g)$ of goal-conditioned policy $\pi(a \mid s, g)$.

**Definition 1** (Combinatorial generalisation). *Let a set of context-conditioned policies $\{\beta_h(a \mid s)\}$ be given, along with a prior over contexts $p(h)$. Let $\beta(a \mid s)$ be the policy constructed via Eq. (5). Let $\pi(a \mid s, g)$ be a policy for evaluation. The combinatorial generalisation of a policy $\pi(a \mid s, g)$ measures the differences in goal-reaching performance for goals sampled $g \sim p_+^\beta(s_{t+} \mid s)$ versus goals sampled from $g \sim \mathbb{E}_{p(h)}[p_+^{\beta_h}(s_{t+} \mid s)]$:*

$$\underbrace{\mathbb{E}_{\substack{s\sim p_+^\beta(s) \\ g\sim p_+^\beta(s_{t+}|s)}} \Big[ f^{\pi(\cdot|s,g)}(s, g) \Big]}_{\textit{test performance}} - \underbrace{\mathbb{E}_{\substack{h\sim p(h),\, s\sim p_+^{\beta_h}(s) \\ g\sim p_+^{\beta_h}(s_{t+}|s)}} \Big[ f^{\pi(\cdot|s,g)}(s, g) \Big]}_{\textit{train performance}}. \tag{8}$$

The precise way performance $f$ is measured is not important for our analysis: "generalisation" simply means that the performance under one distribution is similar to the performance under another. In our experiments, we will look at performance measured by the success rate at reaching the commanded goal. On the surface, it could seem like both the test and train distributions are the same. Lemma 3.1 about reducing mixtures of policies to a single Markovian policy seems to hint that this might be true.

Indeed, this distinction has not been made before while analysing OCBC methods [16, 45]. This misconception is demonstrated by the following lemma:

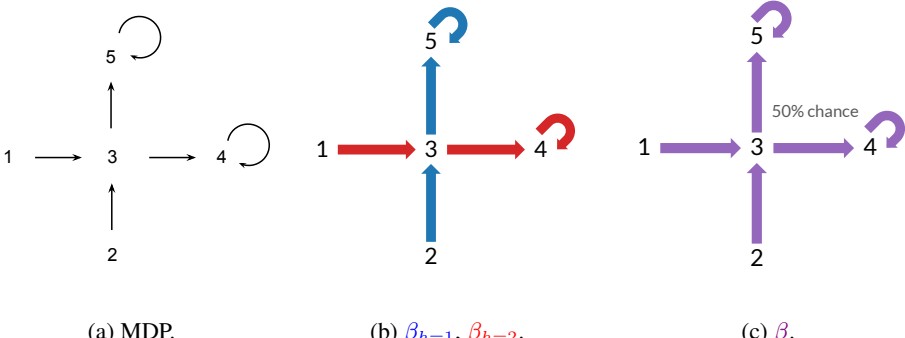

(a) MDP.  (b) $\beta_{h=1}, \beta_{h=2}$.  (c) $\beta$.

Figure 1: *(a)* The MDP has 5 states and two actions (up and right). *(b)* **Training distribution**: Data is collected using two contexts conditioned policies shown in blue and red. *(c)* **Testing distribution**: The behavior cloned policy (equation 5) is shown in purple. During training, the state-goal pair $\{s_t = 2, s_{t+} = 4\}$ is never sampled, as no data collecting policy goes from state 2 to state 4. But the behavior cloned policy has non zero probability of sampling the state-goal pair $\{s_t = 2, s_{t+} = 4\}$. Because of this discrepancy between the train and test distributions, OCBC algorithms do not have any guarantees of outputting the correct action for the state-goal pair $\{s_t = 2, s_{t+} = 4\}$. Whereas dynamic programming based methods can propagate rewards through the backwards *stitched* path of $4 \rightarrow 3 \rightarrow 2$ to output the correct action.

**Lemma 4.1.** *There exist a collection of policies $\{\beta_h\}$ and context distribution $p(h)$ such that, conditioned on a state, the distribution of states and goals for the data collecting policies (training) is different from the distribution of states and goals (testing) for BC policy $\beta$.*

$$\mathbb{E}_{p(h)} \left[ p_+^{\beta_h}(s_{t+} \mid s) p_+^{\beta_h}(s) \right] \neq p_+^{\beta}(s_{t+} \mid s) p_+^{\beta}(s) \quad \text{for some states } s, s_{t+}. \tag{9}$$

*Proof.* The proof is base on a simple counterexample, shown in Fig. 1. See the related caption for a sketch of proof and Appendix D.1 for the formal one. □

In summary, while the BC policy $\beta(a \mid s)$ will visit the same states as the mixture of data collecting policies *on average*, conditioned on some state, the BC policy $\beta(a \mid s)$ may visit a different distribution of future states than the mixture of policies. Even if infinite data is collected from the data collecting policies, there can be pairs of states that will never be visited by any one data collecting policy in a single trajectory. The important implication of this negative result is that stitching requires the OCBC algorithm to recover a distribution over state-goal pairs ($\beta$) which is different from the one it is trained on ($\beta_{h=1}, \beta_{h=2}$).

In theory, the training distribution has enough information to recover the state-goal distribution of the BC policy without the knowledge of the contexts of the data collecting policies. It is upto the algorithm to extract this information. Many RL methods can recover the test distribution implicitly and sidestep this negative result by doing dynamic programming (i.e., temporal difference learning). One way of viewing dynamic programming is that it considers all possible ways of stitching together trajectories, and selects the best among these stitched trajectories. But OCBC algorithms based on SL [1–3, 46, 47] can only have guarantees for iid generalisation [53]. And in line with previous works studying other forms of combinatorial generalisation in SL [10, 54], it is not clear apriori why these methods should have the combinatorial generalisation property, leading to the following hypothesis: *Conditional imitation learning methods do not have the combinatorial generalisation property.*

We will test this hypothesis empirically in our experiments. In Appendix A, we discuss connections between stitching and spurious correlations.

## 5  TEMPORAL AUGMENTATION FACILITATES GENERALISATION

The previous section allows us to rethink the oft-sought "stitching" property as a form of generalisation, and measure that generalisation in the same way we measure generalisation in

SL: by measuring a difference in performance under two objectives. Casting stitching as a form of generalisation allows us to employ a standard tool from SL: data augmentation. When the computer vision expert wants a model that can generalize to random crops, they train their model on randomly-cropped images. Indeed, prior work has applied data augmentation to RL to achieve various notions of generalisation [32, 52]. However, we use a different type of data augmentation to facilitate stitching. In this section, we describe a data augmentation approach that allows OCBC methods to improve their stitching capabilities.

Recall that OCBC policies are trained on $(s, a, g)$ triplets. To perform data augmentation, we will replace $g$ with a different goal $\tilde{g}$. To sample these new goals $\tilde{g}$, we first take the original goal $g$ and identify states from the offline dataset which are nearby to this goal ( Section 5). Let $w$ denote one of these nearby "waypoint" states. Looking at the trajectory that contains $w$, the new goal $\tilde{g}$ is a random state that occurs after $w$ in this trajectory. We visualize this data augmentation in Fig. 2.

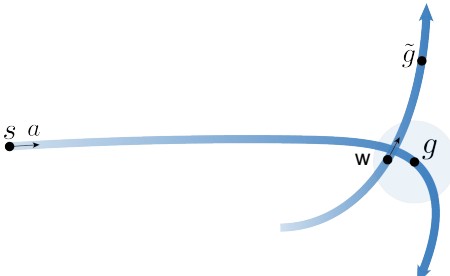

Figure 2:    DATA AUGMENTATION FOR STITCH-ING: After sampling an initial training example $(s, a, g)$ (Eq. (7)), we look for a *waypoint* state $w$ in the light blue region around the original goal $g$, and then sample a new *augmented* goal $\tilde{g}$ from later in that trajectory. This is a simple approach to sample cross trajectory goals $\tilde{g}$ such that the action $a$ is still an optimal action at state $s$.

**Identifying nearby states.** The problem of sampling nearby states can be solved by clustering all the states from the offline dataset before training. This assumes a distance metric in the state space. Using this distance metric, every state can be assigned a discrete label from $k$ different categories using a clustering algorithm [55, 56]. Although finding a good distance metric is difficult in high-dimensional settings [57], our experiments show that using a simple L2 distance leads to significant improvement, even for image-based tasks.

**Method summary.** Algorithm 1 summarizes how our data augmentation can be applied on top of existing OCBC algorithms. Given a method to group states, we can add our data-augmentation to existing OCBC algorithms in about 5 lines of code (marked in blue). In our experiments, we use the k-means algorithm [55]. To sample nearby waypoint states, we randomly sample a state from the same group (same cluster) as the original goal. The augmented goal is then sampled from the future of this waypoint (See  Fig. 2).

---

**Algorithm 1** Outcome-conditioned behavioral cloning with (temporal) data augmentation. The key contribution of our paper is this form of data augmentation, which is shown in blue text.

---
1: **Input**: Dataset : $D = (\{s_0, a_0, \dots\})$.
2: Initialize OCBC policy $\pi_\theta(a|s, g)$ with parameters $\theta$.
3: Set $\epsilon$ = augmentation probability, $m$ = mini-batch size.
4:  $(\{d_0, d_1, \dots\}) = \text{CLUSTER}(\{s_0, s_1, \dots\})$.                              ▷ Group all states in the dataset.
5: **while** not converged **do**
6:     **for** $t = 1, \cdots, m$ **do**
7:         Sample $(s_t, a_t, g_{t+}) \sim D$.                              ▷ Equation (7)
8:         With probability $\epsilon$ :
9:             Get the group of the goal: $k = d_{t+}$.
10:            Sample waypoint states from the same group: $w \sim \{s_i; \forall i \text{ such that } d_i = k\}$ .
11:            Sample augmented goal $\tilde{g}$ from the future of $w$, from the same trajectory as $w$.
12:            Augment the goal $g_{t+} = \tilde{g}$.
13:        Collect the loss $\mathcal{L}_t(\theta) = -\log \pi_\theta(a_t \mid s_t, g_{t+})$.
14:     Update $\theta$ using gradient descent on the mini-batch loss $\frac{1}{m} \sum_{t=1}^{m} \mathcal{L}_t(\theta)$
15: **Return :** $\pi_\theta(a|s, g)$

---

**Theoretical intuition on temporal data augmentation.**    While data augmentations in general do not have exact theoretical guarantees, we can prove that temporal data augmentation, under certain

smoothness assumptions, will generate additional state-goal pairs which may not be seen otherwise during training. In Appendix D.2, we show that there exists a hierarchy of distributions with increasing stitching abilities, where 0-step distribution corresponds to the train distribution (Eq. (9), left) and the per-step distribution corresponds to the test distribution (Eq. (9), right). We prove that applying temporal data augmentation once, samples state-goals from the 1-step distribution.

**Lemma 5.1.** *Under the smoothness assumptions mentioned in Appendix D.2 (Eq. (11)), for all $s, a$ pairs, temporal data augmentation $p^{temp\text{-}aug}(g \mid s, a)$ approximately samples goal according the distribution of one-step stitching policy ($p^{1\text{-}step}(g \mid s, a)$).*

Intuitively, the smoothness assumptions are required to ensure that nearby states have similar probabilities under the data collection policies. In Fig. 2 for example, this ensures taking action $a$ from state $s$ has similar probabilities of reaching nearby states $w$ and $g$. For the complete proof as well as more details see Appendix D.2.

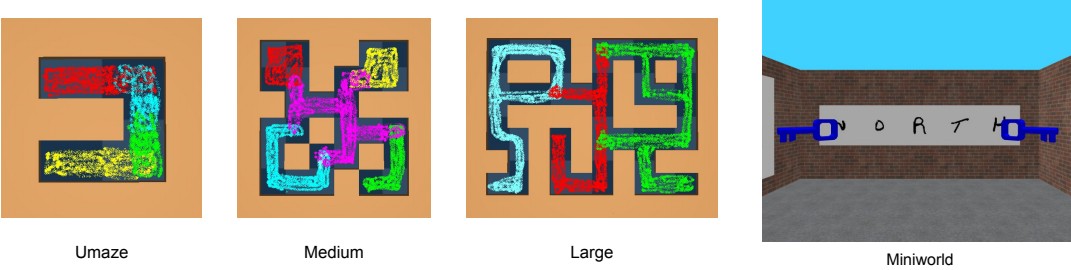

Umaze          Medium          Large                  Miniworld

Figure 3: **Goal conditioned RL :** Different colors represent the navigation regions of different data collecting policies. During data collection, these policies navigate between random state-goal pairs chosen from their region of navigation. These visualisations are for the "point" mazes. The "ant" maze datasets are similar. Appendix Fig. 12 shows the "ant" maze datasets.

Figure 4: **Return conditioned RL :** We visualise our new image based and partially observable environment created using Miniworld [58].

## 6 EXPERIMENTS

The experiments aim *(1)* to verify our theoretical claim that OCBC methods do not always exhibit combinatorial generalisation, even with larger datasets or larger transformer models, and *(2)* to evaluate how adding temporal augmentation to OCBC methods can improve stitching in both state-based and image-based tasks. All experiments are conducted across five random seeds.

**OCBC methods.** RvS [3] is an OCBC algorithm that uses a fully connected policy network and often achieves results better than TD-learning algorithms on various offline RL benchmarks [3]. DT [2] treats RL as a sequential SL problem and uses the transformer architecture as a policy. DT outputs an action, conditioning not only on the current state, but a history of states, actions and goals. See Appendix C.2 for implementation details.

### 6.1 TESTING THE ABILITY OF OCBC ALGORITHMS AND TEMPORAL DATA AUGMENTATION TO PERFORM STITCHING.

While the maze datasets from D4RL [15] were originally motivated to test the stitching capabilities of RL algorithms, we find that most test state-goal pairs are already in the training distribution. Thus, a good success rate on these datasets does not necessarily require stitching. This may explain why OCBC methods have achieved excellent results on these tasks [3], despite the fact that our theory suggests that these methods do not perform stitching. In our experiments, we collect new offline datasets that precisely test for stitching (see Fig. 3 and Fig. 12 for visualisation). To collect our datasets, we use the same "point" and "ant" mazes (umaze, medium and large) from D4RL [15]. To test for stitching, we condition OCBC policies to navigate between (state, goal) pairs previously unseen *together*, and measure the success rate. In Fig. 3, this conditioning corresponds to (state, goal) pairs that appear in differently coloured regions. Each task consists of 2-6 randomly chosen (state, goal) pairs from different regions in the maze. In Appendix C.3, we discuss the important differences between the D4RL and our datasets, which are necessary to test for stitching.

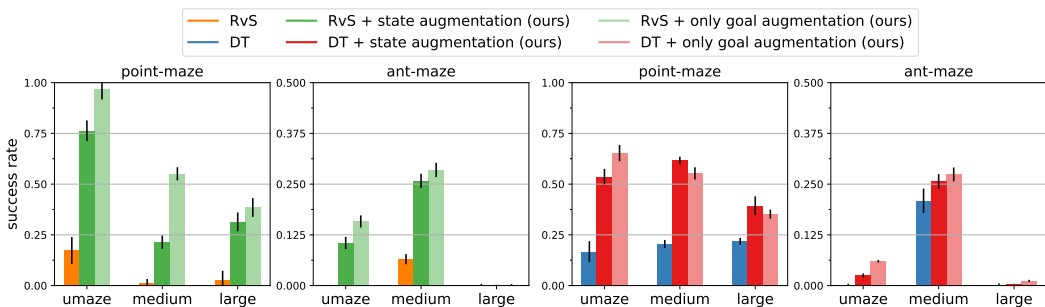

Figure 5: **Adding data augmentation outperforms the OCBC baselines on most tasks.** "Only goal augmentation" refers to an oracle version of our augmentation that uses privileged information $(x, y$ coordinates) when performing augmentation. Adding temporal data augmentation (both standard and oracle versions) improves the performance of both RvS and DT on $5/6$ tasks.

**Results.** In Fig. 5, we can see that both DT and RvS struggle to solve unseen tasks at test-time. However, applying temporal data-augmentation to RvS improves the goal-reaching success rate on $5/6$ tasks, because the augmentation results in sampling (state, goal) pairs otherwise *unseen together*. To show that temporal data augmentation can also be applied to only *important* parts of the state, based on extra domain knowledge, we also compare an oracle version of our data augmentation. This oracle version uses only the $x, y$ coordinates from the state vector to apply the K-means algorithm. Figure 5 also shows that using extra domain knowledge can further improve performance.

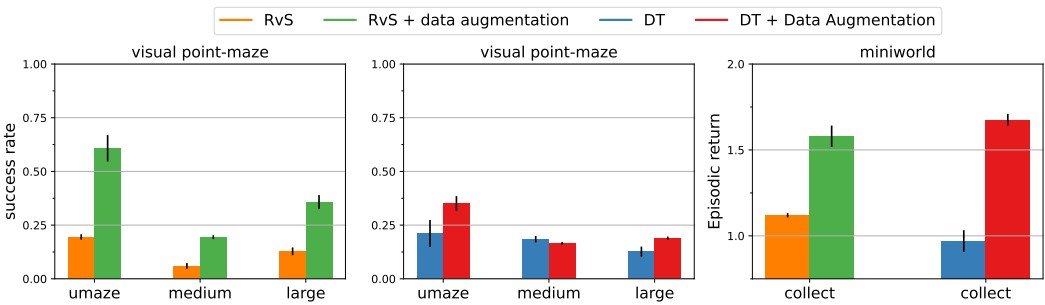

Figure 6: **Temporal data augmentation on image-based tasks.** It is difficult to find a reliable metric to apply temporal data augmentation in high-dimensional tasks. We show that using a simple L2 distance metric can surprisingly improve the combinatorial generalisation of OCBC algorithms on both goal-conditioned (left and center) and return-conditioned (right) tasks.

## 6.2 CAN TEMPORAL DATA AUGMENTATION WORK FOR HIGH DIMENSIONAL TASKS?

As mentioned in Section 5, it can be difficult to provide a good distance metric, especially for tasks with high-dimensional states. Although this is a limitation, we show that temporal data augmentation, using a simple L2 distance metric, can improve the combinatorial generalization of OCBC algorithms even on high-dimensional image-based tasks. To evaluate this, we use both image-based goal-conditioned and return-conditioned tasks. For the goal-conditioned tasks, we use an image-based version of the "point" mazes 3. The agent is given a top-down view of the maze to infer its location. For the return conditioned tasks, we create a new task using Miniworld [58] (See Fig. 4) called "collect". The task is to collect both the keys and return to the start position. A reward of 1 is received after collecting each key. There are two data collecting policies, each collecting only one key. At test time, the OCBC policy is conditioned on the unseen return of 2 (collect both keys).

**Results.** In Fig. 6, we can see that temporal data augmentation improves the performance of RvS and DT on $4/4$ and $3/4$ tasks, respectively. Although temporal data augmentation can be successfully applied to some high dimensional tasks, it is not guaranteed to succeed 5.1. There remains room for other scalable and robust methods to achieve even better performance.

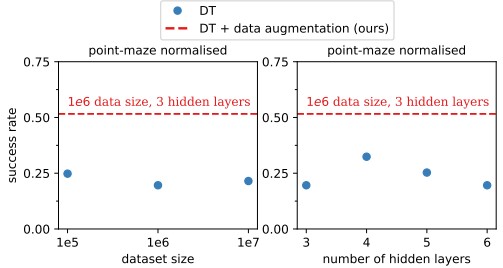
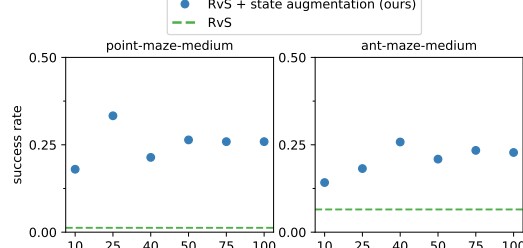

Figure 7: Performance of DT trained on different offline dataset sizes (left) and using a different number of hidden layers (right) averaged across all "point" mazes. Even with larger datasets or models, the generalisation of DT is worse than DT + data augmentation.

Figure 8: Comparison of our data augmentation trained with different numbers of centroids in the K-means algorithm on "point" maze-medium and "ant" maze-medium. Temporal data augmentation corresponding to all values of $k$ outperforms DT on both tasks.

## 6.3 ABLATION EXPERIMENTS.

**Does more data remove the need for augmentation?** Although our theory (Lemma 4.1) suggests that generalisation is required because of a change in distribution and is not a problem due to limited data, conventional wisdom says that larger datasets generally result in better generalisation. To empirically test whether this is the case, we train DT on 10 million transitions (10 times more than Fig. 5) on all "point" maze tasks. In Fig. 7 (left), we see that even with more data, the combinatorial generalisation of DT does not improve much. Lastly, scaling the size of transformer models [59] is known to perform better in many SL problems. To understand whether this can have an effect on stitching capabilities, we increased the number of layers in the original DT model. In Fig. 7 (right), we can see that increasing the number of layers does not have an effect on DT's stitching capabilities.

**How sensitive is temporal data augmentation to the number of centroids used for K-means?** In Fig. 8, we ablate the choice of the number of centroids used in K-means on two environments – "point" maze-medium and "ant" maze-medium. All choices of centroids significantly outperform the RvS method on both tasks.

**Combinatorial generalisation due to spurious relations.** In most of our experiments, OCBC algorithms do exhibit, albeit very low, combinatorial generalisation. We believe this occurs not due to the combinatorial generalisation ability of OCBC algorithms, but due to certain spurious relations that are present in the dataset. In Appendix A, we discuss the relation of combinatorial generalisation with spurious relations. In Appendix B, we perform didactic experiments to show that combinatorial generalisation in OCBC algorithms occurs because the OCBC policy network picks up on such spurious relations.

## 7 DISCUSSION

In this work, we shed light on an area that the community has been investigating recently, *can SL-based approaches perform stitching*. We formally show that stitching requires combinatorial generalisation, and recent SL approaches to RL (OCBC methods) generally do not have any guarantees to perform such generalisation. We empirically verify this on many state-based and image-based tasks. We also propose a type of temporal data augmentation to perform the desired type of combinatorial generalisation precisely and help bridge the gap between OCBC and temporal difference algorithms.

**Limitations.** Our proposed augmentation assumes access to a local distance metric in the state space, which can be difficult to obtain in general. Lifting this assumption and developing scalable OCBC algorithms that generalise is a promising direction for future work.

Overall, our work hints that current SL approaches may not efficiently use sequential data found in RL : even when trained on vast quantities of data, these approaches do not perform combinatorial generalisation (stitching). Due to the temporal nature of RL, it is possible to solve a combinatorial number of tasks from the same sequential data. Similar gains in data efficiency can be made by designing algorithms capable of combinatorial generalisation in other problems involving time series data, for example, audio, videos, and text.

**Acknowledgements.** This work was supported by Mila IDT, Compute Canada, and CIFAR. We thank Artem Zholus, Arnav Jain, and Tianwei Ni for reviewing an earlier draft of our paper. We thank Seohong Park and Mikail Khona for helpful pointers related to the code. We thank Siddarth Venkatraman and members of the Robotics and Embodied AI (REAL) Lab for fruitful discussions throughout the project.

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

## A    RELATIONSHIP WITH SPURIOUS CORRELATIONS.

Handling stitching is somewhat akin to handling spurious correlations studied in the SL community. In the RL setting, we want to navigate from A to B given a dataset that contains some trajectories with A and some with B but none with both A and B. This is somewhat analogous to common settings in object detection in computer vision, where the object background is highly indicative of the object class. For example, the common waterbirds dataset [61] aims to classify images of birds into two classes, "land birds" and "water birds," but the image backgrounds are correlated with the class: water birds are usually depicted on top of a background with water. For evaluation, the classifier is shown an image of a "water bird" on top of a land background (and vice versa). Similar to the RL setting, SL evaluation is done using pairs of inputs that are rarely seen together during training. However, whereas the SL setting aims to learn a classifier that ignores certain aspects of the input, the RL setting is different because the aim is to learn a policy that can reason about both inputs.

There is another connection between the RL setting and spurious correlations, a connection that makes the RL setting look the opposite of the SL setting. For some goal-conditioned RL datasets, the current state is sufficient for predicting which action leads to the goal – the policy does not need to look at the goal. In other datasets, the goal is sufficient for predicting the correct action. However, for navigating between pairs of states unseen together in the dataset, a policy must look at both the state and the goal inputs.

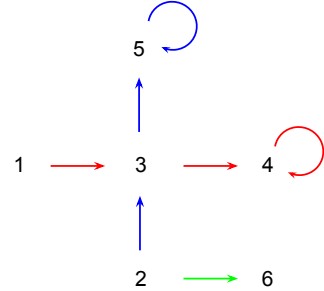

Figure 9: SPURIOUS CORRELATIONS: To understand how spurious correlations are related to stitching, let's look at this simple determinisitc MDP in which three data collecting policies (red, blue and green) collect the offline dataset. Note that an OCBC algorithm which ignores the state, can also achieve a zero training loss Eq. (6) on this offline dataset. Whenever 4 is the desired goal in the dataset, action right is always optimal irrespective of the current state. Any SL algorithm that learns a minimal decision rule will in fact learn to ignore the state to reduce the training loss to zero in this case [60]. But during test time, starting at state 2 and conditioned on goal 4 such an SL algorithm will ignore the current state and move towards right which is clearly suboptimal.

## B    DIDACTIC EXPERIMENTS.

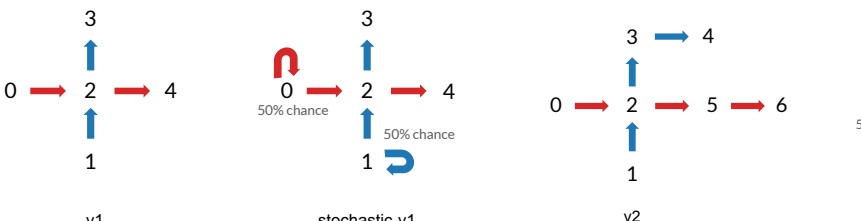

Figure 10:  MDP with 5 states and 2 actions (up and right). All episodes end after taking two actions. Data is collected using two policies (red and blue). The only difference between v1 and v1-stochastic, is the data collecting policies are stochastic at states 0 and 1. The stitching task is to navigate from states 1 to 4.

Figure 11: MDP with 7 states and 2 actions (up and right). All episodes end after taking three actions. Data is collected using two policies (red and blue). The only difference between v2 and v2-stochastic, is the data collecting policies are stochastic at states 0 and 1. The stitching task is to navigate from states 0 to 4.

**Can OCBC algorithms exhibit combinatorial generalisation?**    In Fig. 5, we can see that OCBC algorithms do not always achieve zero performance. To understand why OCBC algorithms seldom perform some amount of stitching, we use two offline datasets collected from a simple didactic MDP (see Fig. 10). In v1 , $3/10$ random seeds of DT are successfully able to navigate from states 1 to 4, while $7/10$ fail. We show that in the both cases, the DT model picks up on one of two spurious

relations present in the dataset. In the *success* seeds, the DT model learns to output action up and ignore the goal, whenever it is in state 1. In the *failure* seeds, the DT model learns to output action right, and ignore the state, whenever the goal is 4. In stochastic-v1, we deliberately remove the spurious relation that leads to success – the model can no longer ignore the goal, as the optimal action depends on it, i.e., if the goal is 1, then the optimal action is right , and if the goal is 3, the optimal action is up. In stochastic-v1, we see that all 10 seeds of DT fail. To be certain that models ignore either states or goals, we check that the model's outputs remain invariant them. This affirms the hypothesis that OCBC algorithms can succeed if the spurious relations in the offline datasets set them up for success.

Similar to the above experiment, we perform more experiments on two offline datasets v2 and stochastic-v2 (see Fig. 11). In v2, $10/10$ random seeds of DT are able to successfully navigate from states 0 to 4. The DT model in all cases picks up on the only spurious relation present in v2 – the optimal action depends only on the current state and the goal can be ignored. We ensure that the model actually ignores the goal at state 0, by checking that its outputs remain invariant to changing goals. In stochastic-v2, we remove this spurious relation; the optimal action at state 0 depends on the goal as well, i.e., if the goal is 0, then the optimal action is up, and if the goal is 6, the optimal action is right. After removing the *success* spurious relation, we observe that the success of DT drops to $0/10$ seeds. This result also algins with the same hypothesis that OCBC algorithms can succeed if the spurious relations in the offline datasets set them up for success at test time.

## C  EXPERIMENTAL DETAILS

### C.1  ENVIRONMENTS

**Goal conditioned environments.**  We use the "point" and "ant" mazes (umaze, medium and large) from D4RL [15]. As discussed in Section 6, we carefully collect our new offline datasets to test for stitching combinatorial (see Fig. 3 for visualisation). In the "point" maze, the task is to navigate a ball with 2 degrees of freedom that is force-actuated in the cartesian directions x and y. In the "ant" maze task, the agent is a 3-d ant from Farama Foundation [62]. To collect data for the "point" maze, we use a PID controller. To collect data for the "ant" maze, we use the same pre-trained policy from D4RL [15]. In Fig. 12, we provide a visualisation of the offline dataset in all "ant" mazes.

**Return conditioned environments.**  For the return conditioned tasks, we create a new task using Miniworld [58] (See Fig. 4). The task is to collect both the keys and return to the start position. A reward of 1 is received after collecting each key. This task is image based and partially observable. The agent recieves a first person view of the world. At any time, it can choose amongst 5 actions – {forward, backward, turn right, turn left, pickup}. There are two data collecting policies, each collecting only one of the key. These data collection policies are implemented by controlling the agent manually.

### C.2  IMPLEMENTATION DETAILS

In this section we provide all the implementation details as well as hyper-parameters used for all the algorithms in our experiments – DT, RvS, and RvS + temporal data augmentation.

**DT.**  We used the exact same hyper-parameters that the original DT paper [2] used for their mujoco experiments. The original DT paper [2] conditioned the transformer on future returns rather than future goals. For our experiments, we switch this conditioning to goals instead. At every time-step the transformer takes in as input the previous action, current state, and desired goal. The desired goal remains constant throughout the episode, but is still fed to the transformer at every timestep. All hyperparameters used for DT are mentioned in Table 1.

**RvS.**  RvS is an OCBC algorithm which uses a fully connected neural network policy. We use the hyperparameters as prescribed by the original paper [3]. All hyperparameters used for RvS are mentioned in Table 2.

**RvS + temporal data augmentation.** As mentioned in Algorithm 1, given a method to cluster states together, it only requires 5 lines of code to add the temporal data augmentation on top of an OCBC method. We use the k-means algorithm from scikit-learn [63] with the default parameters to group states together. Adding data augmentation on top of RvS introduces 2 extra hyperparameters, which we mention in Table 3. We *do not* tune both of these hyperparameters in our paper. Nevertheless, we do ablate the choice of $K$ in k-means.

Table 1: Hyperparameters for DT.

| hyperparameter | value |
|---|---|
| training steps | $2 \times 10^5$ |
| batch size | 256 |
| context len | 5 |
| optimizer | AdamW |
| learning rate | $1 \times 10^{-3}$ |
| warmup steps | 5000 |
| weight decay | $1 \times 10^{-4}$ |
| dropout | 0.1 |
| hidden layers (self attention layers) | 3 |
| embedding dimension | 128 |
| number of attention heads | 1 |

Table 2: Hyperparameters for RvS.

| hyperparameter | value |
|---|---|
| training steps | $1 \times 10^6$ |
| batch size | 256 |
| optimizer | Adam |
| learning rate | $1 \times 10^{-3}$ |
| hidden layers | 2 |
| hidden layer dimension | 1024 |

Table 3: Hyperparameters for temporal data augmentation.

| hyperparameter | value |
|---|---|
| K (number of clusters for k-means) : | |
|     umaze | 20 |
|     medium | 40 |
|     large | 80 |
| $\epsilon$ (probability of augmenting a goal) | 0.5 |

## C.3 DIFFERENCES BETWEEN THE ORIGINAL D4RL AND OUR DATASETS.

We made two main changes in the way our datasets (Fig. 3, Fig. 12) were collected compared to the original D4RL datasets (Fig. 13). *First*, we ensure that different data collecting policies have distinct navigation regions, with only a small overlapping region. This change helps to clearly distinguish between algorithms that can and cannot perform combinatorial generalisation. *Second*, the agent in the original D4RL datasets often moves in a direction that is largely dependent on its current location in the maze. For example in the topmost row of the umaze, the D4RL policy always moves towards the right. To reduce such spurious relations, we randomize the start-state and goal sampling, for the data collecting policies. That is, in the topmost row of our umaze datasets, the data collecting policy moves both towards the right or left, depending on its start-state and goal. Details about how such spurious relations can hamper combinatorial generalisation are discussed in Appendix A.

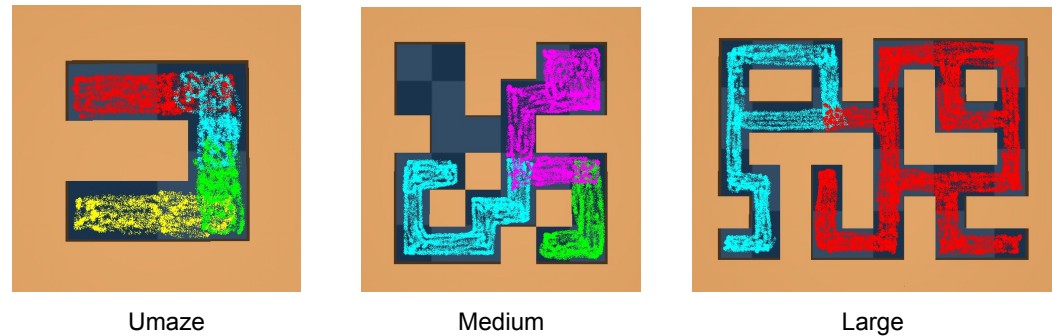


Umaze      Medium      Large


Figure 12: Offline datasets that we collect for the "ant" mazes. Different colors represent the navigation regions of different data collecting policies. See Fig. 3 for a similar visualisation of the "point" maze datasets.

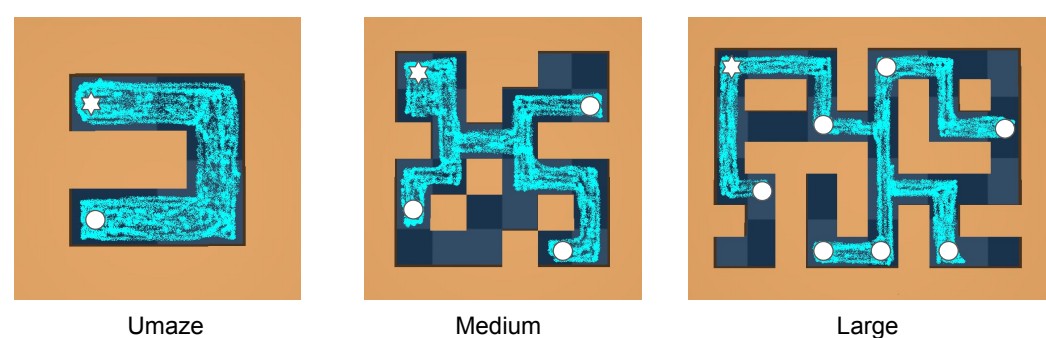


Umaze      Medium      Large


Figure 13: Similar to Fig. 3 and Fig. 12, we create a visualisation of the original d4rl dataset. This is not an exact visualisation of the actual trajectories that are present inside those datasets, but a visualisation of the data collecting policies that those datasets used [15]. During data collection, the policy starts from one starting region, which is marked by a white star. The policy navigates to a goal selected from one of the goal regions, which are marked by white circles. During test time, start-states and goals are selected from similar starting and goal regions, making it difficult to evaluate the combinatorial generalisation of offline RL algorithms.

# D    PROOFS

## D.1    PROOF OF LEMMA 4.1

We prove this Lemma by providing a simple counterexample. Consider the deterministic MDP shown in Figure 1 which has five states $[1, 5]$ and two actions {right, up}. The states four and five are absorbing states; once the agent enters one of these states, it will stay there for eternity. There are two data collecting agents $\beta_{h=1}$ and $\beta_{h=2}$, which navigate upward from state two to state five, and rightward from state one to state four respectively. Both policies collect equal amount of data $p(h = 1) = p(h = 2) = 0.5$. The BC policy $\beta$ (equation 5) is shown in purple to indicate that it is obtained from combining data from both blue and red policies. We will prove that the training and testing distribution (LHS and RHS of Eq. (9)) are not equal for the counterexample state-goal pair $\{s_t = 2, s_{t+} = 4\}$.

$$\mathbb{E}_{p(h)}\left[p_+^{\beta_h}(s_{t+}=4\mid s_t=2)p_+^{\beta_h}(s_t=2)\right] \qquad\qquad p_+^{\beta}(s_{t+}=4\mid s_t=2)p_+^{\beta}(s_t=2)$$

$$=\frac{p_+^{\beta_{h=1}}(4\mid 2)p_+^{\beta_{h=1}}(2)}{2}+\frac{p_+^{\beta_{h=2}}(s_{t+}=4,s_t=2)}{2} \qquad =\frac{p_+^{\beta}(s_{t+}=4\mid s=2)(1-\gamma)}{2}$$

$$=\frac{p_+^{\beta_{h=1}}(4\mid 2)(1-\gamma)}{2}+\frac{0}{2} \qquad\qquad =\frac{(1-\gamma)^2}{2}\sum_{t=0}^{\infty}\gamma^t p_t^{\beta}(4\mid 2)$$

$$=\frac{(1-\gamma)^2}{2}\sum_{t=0}^{\infty}\gamma^t p_t^{\beta_{h=1}}(4\mid 2) \qquad\qquad =\frac{(1-\gamma)^2}{2}(\frac{\gamma^2}{2}+\frac{\gamma^3}{2}+\dots)$$

$$=\frac{(1-\gamma)^2}{2}\times 0=0 \qquad\qquad =\frac{(1-\gamma)\gamma^2}{4}$$

The LHS and RHS are unequal for all values of $\gamma\in(0,1)$.

## D.2 TEMPORAL DATA AUGMENTATION DOES APPROXIMATE ONE-STEP STITCHING.

**Intuition.** The reason why OCBC sampling does not lead to stitching is that it samples a context $h\sim p(h\mid s)$ at the start state, and commits to it by following $\beta_h$ for the entire trajectory. On the other hand, the BC policy follows an average of all behaviour policies at every step of trajectory $(\sum_h \beta_h(a\mid s)p^{\beta}(h\mid s))$. One could think of the BC policy, as sampling a new context $h\sim p(h)$ at every timestep of the trajectory, and following the corresponding $\beta_h$ for that timestep. Intuitively, this ensures that all possible *combinations* of behaviors are sampled by the BC policy. These two ways of sampling trajectories : (1) sampling a context only once at the start and committing to it or (2) sampling a new context at every timestep, are two extremes, and one can think of intermediate ways of sampling trajectories. For example, sample a context $h\sim p(h)$ and a time duration $t\sim p(t)$, and follow $\beta_h$ for timesteps "$t$", and then sample a new context $\tilde{h}\sim p(h)$ and follow $\beta_{\tilde{h}}$ for the rest of the trajectory. Since this form of sampling trajectories changes context only once per trajectory, we call the resulting non-stationary policy as one-step stitching policy. If we do this process twice per trajectory, the resulting non-stationary policy is called two-step stitching policy. According to this notation, the BC policy can be thought of as per-step stitching policy[3]. We will prove that for any state-action pair, the support of the goals reached by $n$-step stitching policies as $n$ increases, is non decreasing. Hence, sampling from the distribution of $n$-step stitching policies with a larger $n$, can sample unseen combinations of state-goal pairs. Finally, we will prove that applying data-augmentation once, under smoothness assumptions, is approximating the distribution of the one-step stitching policy.

**N-step stitching policy.** Before moving to the proofs, we need to define the distribution of future states sampled from the $n$-step stitching policy. Let $p(t)$ be the duration sampling distribution which samples the timestep at which to change the context. We assume that $p(t)$ has non zero support for all time-steps ($\{0,1,\dots\}$). In our case, $p(t)$ is the geometric distribution (geom$(1-\gamma)$). Hence, for a state-action pair $s,a$, the distribution over goals reached by the $n$-step stitching policy is

$$p^{\text{n-step}}(g\mid s,a)$$
$$=\int_{w_{1:n}}\sum_{h_1}p(h_1\mid s,a)p_+^{\beta_{h_1}}(w_1\mid s,a)\Pi_{i=2}^{n+1}\sum_{h_i}p(h_i\mid w_{i-1})p_+^{\beta_{h_i}}(w_i\mid w_{i-1})dw_{1:n}$$

Here $w_i$ is the sub-goal at which the $i^{th}$ switch is made and $w_{n+1}$ is defined as $g$.

**Lemma D.1.** *For $n\in\mathbb{N}$, for all $s,a$, the support of $p^{n+1\text{-step}}(g\mid s,a)$ is at least equal to the support of $p^{n\text{-step}}(g\mid s,a)$, assuming that the duration sampling distribution $p(t)$ has non zero support for all timesteps (for example, geom$(1-\gamma)$).*

---

[3]The BC policy is a stationary policy, as it samples a new contexts at every timestep. There is no dependency on time.

*Proof.* Assume that a goal-state $g$ belongs to $\text{Supp}\, p^{\text{n-step}}(g \mid s, a)$. We will prove that this implies that $g \in \text{Supp}\, p^{\text{n+1-step}}(g \mid s, a)$. Because $g \in \text{Supp}\, p^{\text{n-step}}(g \mid s, a)$, we can assume without loss of generality that $g$ is visited $k$ time-steps after switching the context $n$ times.

Before switching for $n+1^{th}$ time, the distribution of states visited by the $n+1$-step policy is identical to the distribution of $n$-step policy. The $n + 1^{th}$ switching duration is drawn from $t_{n+1} \sim p(t)$. Because $p(t)$ has support over all timesteps ($\text{Supp}\, p(t) = \{0 + \mathbb{N}\}$), we know that $p(t_{n+1} > k) > 0$. That is, the probability that $n+1$-step policy does not make the $n+1^{th}$ switch for atleast $k$ timesteps is non zero. This proves that for any finite value of $k$, there is a non zero probability that the distribution of states visited by the $n + 1$-step policy is identical to the $n$-step policy for atleast $k$ timesteps after the $n^{th}$ switch. Hence the goal state $g$ can also be visited by $n + 1$-step policy. Hence proved that $\text{Supp}\, p^{\text{n-step}}(g \mid s, a) \subseteq \text{Supp}\, p^{\text{n+1-step}}(g \mid s, a)$. $\square$

Using the above lemma, we can say that

$$\text{Supp}\, p^{\text{n-step}}(g \mid s, a) \subseteq \text{Supp}\, p^{\text{n+1-step}}(g \mid s, a) \subseteq \text{Supp}\, p^{\text{n+2-step}}(g \mid s, a) \ldots \tag{10}$$

Hence proved that for any state-action pair, the support of the goals reached by n-step stitching policies as n increases, is non decreasing. And from 4.1, we know that there can be states which are only visited by $\{n > 0\}$-step stitching policies. Hence, it is possible that the relations in Eq. (10) are strict.

**Temporal data augmentation approximates one-step stitching policy.** Under smoothness assumptions for the discounted state occupancy distribution of the data collecting policies, we can show that temporal data augmentation Fig. 2 approximates the distribution of one-step stitching policy. Specifically, we assume that for all $s, a, g$ pairs and all data collecting policies $\beta_h$, $p_+^{\beta_h}(g \mid s, a)$ is $L$ Lipschitz

$$|p_+^{\beta_h}(g \mid s, a) - p_+^{\beta_h}(w \mid s, a)| \leq L(||g - w||_2) \tag{11}$$

This is an important assumption for temporal data augmentation as it ensures that states which are close together have similar reachability. Prior work has also studied this assumption [64–66] and applied it to practical settings [67]. Finally, our theoretical analysis uses a form of temporal data augmentation which clusters states only with their nearest neighbour. That is, each group in Algorithm 1 contains atmost 2 unique states.

**Lemma D.2.** *Under the smoothness assumptions above, for all $s, a$ pairs, temporal data augmentation $p^{temp\text{-}aug}(g \mid s, a)$ approximately samples goal according the distribution of one-step stitching policy ($p^{1\text{-}step}(g \mid s, a)$).*

*Proof.*
$p^{\text{temp-aug}}(g \mid s, a)$

$$\overset{a}{=} \int_w \sum_h p(h \mid s, a)(p_+^{\text{temp-aug}}(w \mid s, a)) \sum_{\tilde{h}} p(\tilde{h} \mid w) p_+^{\beta_{\tilde{h}}}(g \mid w) dw$$

$$\overset{b}{=} \int_{w_1} \sum_h p(h \mid s, a)(p_+^{\beta_h}(w_1 \mid s, a) \pm \epsilon L) \sum_{\tilde{h}} p(\tilde{h} \mid w_1) p_+^{\beta_{\tilde{h}}}(g \mid w_1) dw_1$$

$$= \int_{w_1} \sum_h p(h \mid s, a) p_+^{\beta_h}(w_1 \mid s, a) \sum_{\tilde{h}} p(\tilde{h} \mid w_1) p_+^{\beta_{\tilde{h}}}(g \mid w_1) dw_1 \pm \epsilon L \int_{w_1} \sum_{\tilde{h}} p(\tilde{h} \mid w_1) p_+^{\beta_{\tilde{h}}}(g \mid w_1) dw_1$$

$$= \int_{w_1} \sum_h p(h \mid s, a) p_+^{\beta_h}(w_1 \mid s, a) \sum_{\tilde{h}} p(\tilde{h} \mid w_1) p_+^{\beta_{\tilde{h}}}(g \mid w_1) dw_1 \pm \mathcal{O}(\epsilon L)$$

$$= p^{\text{1-step}}(g \mid s, a) \pm \mathcal{O}(\epsilon L)$$

$\square$

*Here $\epsilon$ is the maximum cutoff distance used by the temporal data augmentation to group states together.*

*(a)* is followed by the fact that temporal data augmentation first sample a waypoint $w$, and then sample a future goal $g$ from the trajectory containing $w$. This new trajectory can correspond to any data collecting policy $\beta_h$ with probability $p(\tilde{h} \mid w)$. *(b)* $w_1$ is the initial goal sampled by temporal data augmentation, which is then substituted by its nearest neighbour $\tilde{w}$. It follows from Eq. (11) that $p_+^{\beta_h}(w \mid s, a) - \epsilon L \leq p_+^{\beta_h}(w_1 \mid s, a) \leq p_+^{\beta_h}(w \mid s, a) + \epsilon L$, as epsilon is the maximum possible $L2$ norm between $w_1$ and $w$.

