# OpenReview forum: "Closing the Gap between TD Learning and Supervised Learning - A Generalisation Point of View."
_ICLR.cc/2024/Conference — ICLR 2024 poster_

### Official Review · Reviewer_6ajj · 2023-10-30

**Soundness:** 3 good
**Presentation:** 3 good
**Contribution:** 2 fair
**Rating:** 6
**Confidence:** 3

**Summary:**

* This work discusses a form of generalization in reinforcement learning (RL) called "stitching generalization." According to the paper, the stitching generalization cannot be achieved with existing supervised-learning (SL) based RL algorithms because it is fundamentally different from the generalization on independent & identically distributed (i.i.d.) dataset. This is explained via a concrete example of an MDP depicted in Figure 1.
* Also, this work propose "temporal data augmentation" technique for goal-conditioned RL to actually implement stitching generalization inside the SL-based RL algorithms. This technique is based on careful sampling of augmented goal, which is determined by *closeness" of states based on a clustering algorithm.
* Moreover, this work provides a novel benchmark to evaluate the stitching capability of RL algorithm.
* Lastly, this work empirically show the efficacy of the temporal data augmentation + outcome conditional behavior cloning (OCBC) algorithms on their own benchmark.

**Strengths:**

* The motivation is clear and the main message is well presented. I enjoyed reading the paper.
* Their main method of data augmentation is novel and intuitive.
* The empirical results are promising, even after making the task more difficult to generalize.
* Lastly, the limitations of their work is clearly mentioned. I also look forward to scalable version of their algorithm.

**Weaknesses:**

* **Theoretical contribution is a bit marginal.**
  - The authors claim that they “provide a theoretical framework for studying stitching.” They indeed provided a formal definition of stitching generalization in Definition 1. However, it is not actually used anywhere. If it is not applicable to deduce any theoretical guarantee, what is the purpose of proposing such definition? At least for the counterexample proposed in Lemma 4.1 and depicted in Figure 1, the authors should be able to provide any kinds of implications on the (lower bound of) stitching generalization. This will definitely strengthen the theoretical contribution of the paper. If it is not possible, please kindly explain the reason/context.
* **Isn’t it a problem of trajectory-based sampling rather than transition-based sampling that it is impossible to implement stitching with SL-based methods?**
  - At first glance, I thought that the statement “stitching generalization is not the same as i.i.d. generalization” is not quite the problem of supervised learning itself. Let’s take a look at the counterexample in Figure 1. If we break down the trajectories collected by policies $\beta_{h=1}$ and $\beta_{h=2}$ into transitions, can’t even a supervised method learn something from a virtual trajectory, say, 2-3-4, which was not exactly in the dataset, by only i.i.d. sampling the transitions? To rephrase my question, can’t a SL-based method learn stitched behaviors by transition-based sampling or so-called *bootstrapping*? I don’t think this would be exactly the same as the proposed temporal augmentation method. Please correct me if I’m wrong.
* **Weakness on Algorithm 1 (OCBC + Temporal Data Augmentation) regarding additional hyperparameter**
  - It uses clustering algorithm, so it has an additional hyperparameter, $k$ for k-means algorithm. To the best of my knowledge, the main paper does not discuss on the effect of the choice of such hyperparameter, which might be difficult to properly tune.
* **It seems necessary to conduct additional Experiments on DT + temporal augmentation.**
  - Although the paper empirically proves that it can enhance the data utilization of *RvS* algorithm with their proposed augmentation technique, it does not provide any results on ‘’DT + temporal augmentation’’ combination. This experiment will strengthen the empirical contribution of the paper. If it is not applicable, please kindly explain why.
* **There seems to be some minor typos.**
  - pg 1. Kaelbling (1993) $\rightarrow$ (Kaelbling, 1993)
  - pg 3., Equation (1). $\mathbb{E}\_{s\sim p_0(s_0)} [p_t^{\pi}(s_{t+} =g \mid s_0=s)]$ $\rightarrow$  $\mathbb{E}\_{s\sim p_0(s_0)}[p_+^{\pi}(s_{t+} =g \mid s_0=s)]$
  - pg 3. “… and then sampling a trajectory from the corresponding policy $\\{\beta(a\mid s, h)\\}$.” $\rightarrow$ $\beta(a\mid s, h)$

**Questions:**

* In Definition 1, does the performance function $f$ have any relavance to the objective $J$ defined in Equation (3) or the maximum likelihood objective defined in Equation (6) and (7)?
* In page 5, what does it mean by ‘combinatorial generalization’? Do you have a definition or any reference for that?
* Just curious: regarding Lemma 4.1, do you have any comments or implications on the (sort of) bias term $\E_{p(h)} \left[ p^{\beta_h}_+ (s_{t+} \mid s) p^{\beta_h}_+ (s) \right] - p^{\beta}_+ (s_{t+} \mid s) p^{\beta}_+ (s)$ ?

---

> ### Author Response · Authors · 2023-11-16
> **Response to 6ajj (1 / 2)**
>
> Dear reviewer,
> We thank the reviewer for their detailed feedback. It seems like the reviewer's main concerns are the importance of theoretical contributions, introduction of a new hyperparameter and some missing experiments. We have revised Sections 5, 6 and:
>
> 1) Added Figure 8, in appendix B, to show that data augmentation works for a range of K values {10 25 40 50 75 100}.
>
> 2) Added Figure 11, in Appendix B which shows preliminary results for scalable SL algorithms built on top of our work can also achieve stitching generalization.
>
> 3) Added Appendix D.1, which shows that uses our definition of stitching, to prove that the optimal OCBC performs policy improvement over the behavior policy.
>
> **Do these clarification and new experiments and proofs address all the reviewer's concerns?**
>
> > They indeed provided a formal definition of stitching generalization in Definition 1. However, it is not actually used anywhere.
>
> We believe a formal definition of stitching generalization is useful because we use the definition to generate the datasets used to evaluate stitching. We revise section 5, para “Popular offline datasets do not evaluate stitching.” (see green text and equation), to clarify this. In Appendix D.1, we add a section which shows that the Bayes optimal OCBC policy that is learnt on a “stitched dataset”, where states-goals are sampled using the test distribution, is an improvement over the BC policy. In offline RL, policy improvement guarantees over the behavior policy are important because optimality guarantees are not generally realisable [1,2].
>
> > Weakness on Algorithm 1 (OCBC + Temporal Data Augmentation) regarding additional hyperparameter
>
> In appendix B, Figure 8, we add a new experiment which ablates the value of the number of centroids on two environments (ant maze and point maze medium). We run experiments with K ranging between {10 25 40 50 75 100}. We can see that all values significantly improve the stitching generalization over the baseline.
>
>
> > Isn’t it a problem of trajectory-based sampling rather than transition-based sampling
>
> OCBC algorithms also learn from individual transitions. For example in Figure 1, they will be trained on transitions (2,up,3) and (3,right,4). But they will never sample (2,up,4), which is out of distribution (has 0 support in the training distribution). There is no guarantee that a SL method will perform on points which are strictly out of distribution. But our paper’s main goal is to take initial steps to endow purely SL methods with these desirable properties of stitching-based algorithms like TD, while still retaining their simplicity and scalability.
>
> > It seems necessary to conduct additional Experiments on DT + temporal augmentation.
>
> The reason we do not add data augmentation with DT is that because DT takes in a context of previous states and actions, the data augmentation method should cluster a history of states and actions, which is difficult to do computationally. We have added this reason in our updated paper (see Caption of Figure 4, text in green).
>
> > I also look forward to scalable version of their algorithm.
>
> We believe that purely SL algorithms which learn state representations containing information about future outcomes, can possess desirable properties associated with TD learning while still retaining their simplicity and scalability. In Figure 11, in Appendix B, we include an experiment that shows using such representations as inputs for the OCBC policy, improves stitching generalization without the need of a distance metric.
>
> > There seems to be some minor typos.
>
> Thank you for pointing these out, we have fixed these typos.

---

> > ### Author Response · Authors · 2023-11-16
> > **Response to 6ajj (2 / 2)**
> >
> > Below we will answer your specific questions in detail.
> >
> > > does the performance function any relevance to the objective
> >
> > The performance measure can depend on the practitioner. It can be the time required for the agent to reach the goal (shortest path) or it can be the probability that the agent reaches and stays in the goal (goal reaching). So the objective of OCBC algorithms and the performance measure do not have any dependency between them. The only expectation is that the performance measure should be some meaningful measure of reaching a goal from a state.
> >
> >
> > > what does it mean by ‘combinatorial generalization’?
> >
> > We have added a reference that studies combinatorial generalization in the context of RL. But the similarity is that stitching generalization tests on previously unseen combinations of states and goals which have been separately seen.
> >
> > > do you have any comments or implications on the bias term
> >
> > In this bias term, the marginal distribution over states is the same for both sides (Lemma 3.1). But given any state, the support over the goal space of RHS will always be wider than the support over the goal space of LHS. This can be easily proved. For any given state s,
> > If any of the data collecting policies visits a goal g, this implies that the BC policy will also visit the goal g. By definition the BC policy will imitate all data collecting policies with some non zero probability.
> >
> > But the converse of this is not true in general, that is there will be a goal which the BC policy will visit, but no data collecting policies will visit (lemma 4.1). Hence the implications of this gap are methods that do stitching and can reason / visit a wider distribution over goal from any given state when compared to methods that cannot do stitching.
> >
> > [1] The curse of passive data collection in batch reinforcement learning
> >
> > [2] Offline reinforcement learning: Fundamental barriers for value function approximation

---

> > > ### Author Response · Authors · 2023-11-20
> > > **Rebuttal Follow-up**
> > >
> > > Dear Reviewer,
> > >
> > > We hope that you've had a chance to read our responses and clarification. As the end of the discussion period is approaching, we would greatly appreciate it if you could confirm that our updates have addressed your concerns.

---

### Official Review · Reviewer_4eRX · 2023-11-01

**Soundness:** 2 fair
**Presentation:** 2 fair
**Contribution:** 2 fair
**Rating:** 5
**Confidence:** 4

**Summary:**

This work analyzes SL-based RL approaches for (goal-conditioned) offline RL, where data are collected using various policies and one hopes to "stitch" existing experience/trajectories to generalize to unseen (start, goal) pairs. It shows that common outcome conditional behavioral cloning (OCBC) methods can fail this task and shows a lemma (Lemma 4.1) that learning from training experience with different contexts may not lead to generalization (in terms of the mixed behavior policy). Then the paper proposes to use temporal augmentation that essentially stitches two trajectories together to have better coverage for the (start, goal) pair. Experiments show that the augmentation can be effective for some offline RL tasks.

**Strengths:**

- Studies an important problem of generalization from offline data
- Shows a counterexample of the training and test discrepancy (Lemma 4.1)
- Clear writing in most places

**Weaknesses:**

1. Some key concepts and results are not clearly explained.



    1.1. The core concept, "stitch property", is poorly introduced and readers have to refer to the reference to know what it means. There is no consistent nor clear definition for this "stitch property", making it difficult to understand what the paper is trying to achieve.

    1.2. "Our paper focuses on the problem of stitching generalization, not finding optimal policies; however, stitching generalization should be a necessary component for finding optimal policies." Proof needed.

2. The proposed heuristic of using clustering to stitch close states together lacks theoretical guarantees. What distance metric is suitable here? In the RL context, similar states do not mean they have similar outcomes. For example, two states can be close to each other in Euclidean norm but blocked by a walk and thus not reachable; or two states can be far apart (e.g., top and bottom of a cliff) but one can easily reach the bottom by falling but not the other way around.

3. For the experiment, the paragraph on "Does more data remove the need for augmentation" is not very surprising given the construction of the dataset. In SL, the fact that more data helps generalization is based on the IID assumption, which does not hold in the current setting since the (state, goal) pairs are not seen during training (also Lemma 4.1).

Minor
- $p_+^{\beta_h}(s,a)$ in Eq.(7) is undefined.
- Both generalization and generalisation are used.
- point-maze large taks -> task

**Questions:**

Q1: Any suggestions for a reasonable distance metric for the states?

Q2: Are there any theoretical justification for the temporal augmentation?

---

> ### Author Response · Authors · 2023-11-16
> **Response to 4eRX**
>
> Dear reviewer,
> We thank the reviewer for their detailed feedback. It seems like the reviewer's main concerns are regarding a clear definition of stitching, a theoretical justification for the data-augmentation, and the difficulty of obtaining a good distance metric. We have incorporated these concerns using new updates to the papers and new image based experiments:
>
> 1) We add a new paragraph at the start of Section 4 (green text) to explain how stitching is colloquially associated with different properties of TD learning. We add that we focus on one of these properties as a clear difference between TD and OCBC methods and call it stitching generalization, which we concretely define in Definition 1.
>
> 2) In Figures 9 and 10, in Appendix B, we perform additional experiments with an image-based version of the point mazes. We show that performing data augmentation works with images as well, where a good distance metric is not available.
>
> 3) In section 5, we added a paragraph “Theoretical intuition on temporal data augmentation”, to provide some theoretical justification for data augmentation.
>
> 4) In Figure 11, in Appendix B, we include a preliminary experiment that removes the use of a distance metric, by learning representations. We believe that this approach is a promising direction for the future work that builds on our paper.
>
> Below, we will describe these updates and experiments in more detail and answer the specific reviewer questions **Do these answers, revisions and experiments address all the reviewer's concerns?**
>
> > In the RL context, similar states do not mean they have similar outcomes
>
> We agree with the reviewer that having a good distance metric is challenging and have already acknowledged the limitation in the main paper. But empirically, we show that this limitation might not always matter. We add two sets of experiments ( appendix B.2, Figures 9 and 10 ) on image-based pointmaze tasks and show that K-means using L2 distances between images can perform well, significantly improving over the OCBC baselines on all tasks. We would like to add that for all of our maze experiments, if the cut-off distances for K-means clustering was too large, then it would group together states on different sides of the walls like you suggest. But our empirical results (Figures 4,9, and 10) show that using K-means works well.
>
> > "Does more data remove the need for augmentation" is not very surprising given the construction of the dataset
>
> In this work, we aim to understand and evaluate the stitching properties of OCBC algorithms. The experiments are designed to evaluate this question and support the claim that data augmentation isn't just about inflating the size of the dataset. This claim (and verification) is important because in today's world of large models, many researchers might otherwise be tempted to throw large amounts of data at this problem, without avail.
>
> > Q1: Any suggestions for a reasonable distance metric for the states?
>
> We believe that learning state representations such that the distance between these representations contains information about the similarity of their outcomes, is a scalable and promising direction of future work [1,2]. This will solve the exact problem that you correctly pointed out. In Figure 11, in Appendix B, we include an experiment that shows using such representations as inputs for the OCBC policy, improves stitching generalization without the need of a distance metric.
>
> > Q2: Are there any theoretical justification for the temporal augmentation?
>
> While exact theoretical guarantees for temporal data augmentation will depend on how good the distance metric is, we can think of the temporal data augmentation as trying to diversify the dataset by sampling from $\pi^{\beta}_+(g \mid s)$. Since this is the test distribution for stitching, data augmentation should improve stitching generalization. In section 5, we add a paragraph, “Theoretical intuition on temporal data augmentation” to intuitively explain how data augmentation leads to a better policy.
>
> > Minor
> Thank you for spotting these errors, we have fixed them in our updated paper.
>
> [1] DATA-EFFICIENT REINFORCEMENT LEARNING WITH SELF-PREDICTIVE REPRESENTATIONS
> [2] Contrastive Learning As a Reinforcement Learning Algorithm

---

> > ### Author Response · Authors · 2023-11-20
> > **Rebuttal Follow-up**
> >
> > Dear Reviewer,
> >
> > We hope that you've had a chance to read our responses and clarification. As the end of the discussion period is approaching, we would greatly appreciate it if you could confirm that our updates have addressed your concerns.

---

> > > ### Comment · Reviewer_4eRX · 2023-11-21
> > >
> > > I thank the authors for providing additional explanations and experiments.
> > >
> > > While I do believe the paper studies an important problem, the additional explanation (first paragraph of Section 4) of the stitching property still lacks mathematical rigor. Additionally, the experiment based on images can still be favorable to L2 distance because similar images do have similar meanings in that context.
> > >
> > > Overall, I think the paper has merit in a better understanding of the problem, but the discussion is not mathematically rigorous and the algorithm development lacks theoretical justifications. As a result, I keep my score.

---

> ### Author Response · Authors · 2023-11-22
> **Response to 4eRX**
>
> > the additional explanation (first paragraph of Section 4)
>
> This is meant to be an intuitive introduction on how stitching is colloquially associated with various properties in the literature. In Definition 1, we give the precise definition of stitching generalization. If you think there is something about the definition that isn't rigorous, please let us know. We'd be happy to look into it.
>
> > the algorithm development lacks theoretical justifications
>
> The data augmentation we propose does have theoretical backing. In Figure 2, in the limit of finest clusters, i.e, as the size of the blue circle tends to zero, temporal data augmentation is equivalent to sampling from the Markov chain s -> w -> g. This shows that in the limit of the finest clustering, temporal data augmentation will sample cross trajectory state-goal pairs. In practice, we show that K-means works well across various values of K (See Figure 8).
>
> >  images can still be favorable to L2 distance because similar images
>
> We agree that there can be image based tasks where it will be difficult to make data augmentation work. But the image based maze tasks are challenging, and adding the temporal data-augmentation (5 lines of code) improves the performance of baselines by more than 2 times on all the 3 image based tasks.
>
> **We kindly ask the reviewer to reconsider the paper in light of the revisions and clarifications.**

---

### Official Review · Reviewer_DGi6 · 2023-11-01

**Soundness:** 3 good
**Presentation:** 3 good
**Contribution:** 2 fair
**Rating:** 5
**Confidence:** 3

**Summary:**

The paper explores the differences between dynamic programming-based reinforcement learning (RL) and supervised learning (SL)-based RL. A key property of the former is "stitching" past experiences to address new tasks. The study relates this ability to a unique form of generalization. Through experiments, the authors reveal that SL-based RL methods might lack this stitching capability. To address this, they introduce temporal data augmentation, enabling SL-based RL to handle unseen (state, goal) pairs effectively.

**Strengths:**

1. The paper provides a comprehensive analysis of the differences between dynamic programming-based RL and SL-based RL through the lens of generalization, offering insights into their inherent properties.
2. A novel augmentation method is proposed to make SL more generalizable.

**Weaknesses:**

1. The proposed stitching generalization describes how "far away" different goals are in the sense of transitions, and the proposed method relies on a heuristic to describe that. This is basically what we aim to learn in TD learning and in most cases we cannot just get this kind of information for free. See Q1.
2. The experiments are limited to Antmaze where the proposed clustering would probably work with raw state inputs, but if we imagine pixel inputs, it would be very hard to find some heuristic to make it work, as mentioned in 1.
3. Missing baselines: Contrastive-based learning is also studied for goal-reaching problems as one kind of supervised learning problem, and it might also learn some representation more useful than the baseline included in the paper.

**Questions:**

Q1: The optimal augmentation would be some metric that describes "how "far away" different states are in the sense of transitions" almost perfectly. Then such a metric will basically provide us with the optimal value function. In this sense, getting such a metric could be as hard as solving it with TD, so why does the proposed method necessarily "close the gap" between TD and SL?

---

> ### Author Response · Authors · 2023-11-16
> **Response to DGi6**
>
> Dear reviewer,
> We thank the reviewer for their detailed feedback. It seems like the reviewer's main concerns regarding the assumption of a distance metric, especially in high dimensional states, and the inclusion of contrastive representation-based baselines. We address these concerns by adding two new experiments and updates:
>
> 1) In Figures 9 and 10, in Appendix B, we perform an additional experiment with an image-based version of the point mazes. We show that performing data augmentation using k means on images performs much better than the baseline. Although we agree that finding a good distance metric is difficult, which we have already mentioned in the paper, our new results using images and the existing ant maze tasks with 29 dimensional state spaces, suggests that K-means with simple L2 distance can empirically still perform great.
>
> 2) In Appendix D.2, we prove that contrastive RL will not learn Q values for states, goals that are used to test for stitching.
>
> 3) In Figure 11, in Appendix B, we include a novel version of RvS which takes in input the contrastive representations of states [1]. Just like the reviewer suggested, we see that these representations are indeed able to improve the stitching abilities of OCBC methods. We believe that this approach is a promising direction for the future work that builds on our paper.
>
> Below, we will describe these updates and experiments in more detail and answer the specific reviewer questions **Do these answers, revisions and experiments address all the reviewer's concerns?**
>
> > The proposed stitching generalization describes how "far away" different goals
>
> The information of how far away goals are from a particular state is not directly available in the distance metric. We do not make this assumption, and it is not even required. We only need a distance metric to make sense on very small length scales, that is it should identify states which are very close together. This is easier than describing how “far away” goals are from a given state (Q value of reaching a goal). To actually learn to actually select actions that help you reach unseen goals, the temporal data augmentation we propose is important.
>
> > The experiments are limited to Antmaze
>
> To incorporate your suggestion, we include two sets of experiments with the image based version of point maze tasks (umaze, medium and large). In figure 9 and 10, in appendix B, we see that applying clustering on images and using it for our data augmentation outperforms the baselines on all tasks. It also matches the performance of the state based augmentation in the large maze (Figure 10).
>
> > Missing baselines: Contrastive-based learning is also studied for goal-reaching problems as one kind of supervised learning problem,
>
> In Appendix D.2, we add a proof (based on Lemma 4.1)  which shows that contrastive RL [1] will not estimate the Q function for state-goal pairs used to test for stitching. We will include the actual empirical results in the camera ready version, as we need more time to adapt their code (which is written in a different framework) to our settings.
>
> > it might also learn some representation more useful than the baseline included in the paper.
>
> According to your suggestions, we performed an ablation experiment which uses contrastive representations [1] as inputs to the OCBC policy on point maze tasks (umaze, medium and large). In Figure 11, in Appendix B,  we see that these representations are indeed able to improve the stitching abilities of OCBC algorithms. We have added Appendix B.3, which notes that learning contrastive representations for OCBC algorithms is a promising direction for future work.
>
> > The optimal augmentation would be some metric that describes "how "far away" different states are in the sense of transitions" almost perfectly.
>
> This proposed metric could work well without our augmentation method, however it is more than what is needed to improve performance. We do need a distance metric to cluster close-by states. Intuitively, we want a distance metric that can classify states as very close or not. See for example, figure 2 where the distance metric does not tell us how far $\tilde{g}$ is from $s$. It only tells us that $w$ and $g$ are close-by states.
>
> [1] Contrastive Learning As a Reinforcement Learning Algorithm

---

> > ### Author Response · Authors · 2023-11-20
> > **Rebuttal Follow-up**
> >
> > Dear Reviewer,
> >
> > We hope that you've had a chance to read our responses and clarification. As the end of the discussion period is approaching, we would greatly appreciate it if you could confirm that our updates have addressed your concerns.

---

> > > ### Comment · Reviewer_DGi6 · 2023-11-23
> > >
> > > Thanks for the extra experiments and rebuttals! However I still have concerns about the distance metric: To do clustering as the paper suggests, it is required that the number of steps needed to transit from one state to another in the cluster is small, which means what metric should be used really depends on the underlying dynamics (i.e. the states could be close in the state space but hard to transit in between). Generally speaking, getting such information is not easy and it is hard to imagine some universal way to do it without any stitching, given that how to cluster those nearby states depends on the dynamics. So I will keep my score.

---

### Official Review · Reviewer_dPJE · 2023-11-01

**Soundness:** 2 fair
**Presentation:** 3 good
**Contribution:** 2 fair
**Rating:** 6
**Confidence:** 3

**Summary:**

First, the paper identifies one important capability present in dynamic programming but lacking in supervised learning approaches to RL: the stitching property, which generalizes between different trajectories. - The second conceptual leap connects the stitching property to a certain kind of generalization. Finally, the authors propose to outfit data augmentation from supervised learning for reinforcement learning as a simple method to improve generalization. Experiments on two enviornments show that RvS with state and goal augmentation improve over RvS and DT. It is also claimed that more data did not help the DT improve, nor did increasing the number of layers.

**Strengths:**

- The theoretical foundation developed in this paper is interesting. There is high value in formalizing stitching, and hypothesizing on fundamental limits of certain algorithmic approaches that apply supervised learning to reinforcement learning problems. I also like the ethos of the approach, bringing data augmentation to RL for a specific purpose

**Weaknesses:**

- Some of the theoretical arguments fail to convince. I think there are a few details that seem inconsistent throughout the paper, such as the fact that data is not the fundamental limitation vs popular datasets not evaluating stitching because they include most (s,g) pairs.
- The experimental evaluation seems limited overall, and somewhat preliminary. I usually do not put much weight on this kind of weakness, but I also am not comfortable with the theoretical positioning of the paper. Hence, I must put more weight on the experiments in my decision for this paper.

**Questions:**

- Section 1 (Stitching as goal-state pairs): My understanding of the stitching property is that it is more general: given two partially disjoint trajectories that partially optimal, stitching via dynamic programming can combine the two partially optimal trajectories towards an optimal trajectory. Is your definition equivalent, is it a generalization or is it specific to the goal-directed setting?
- Section 1 (Limited data vs generalization): You state that this is not a problem of limited data, because " there can be (state, goal) pairs that are never visited in the same trajectory, despite being frequented in separate trajectories." In a way this is a source of limited data, if the "data" is a trajectory rather than an experience tuple. I think this is implicit in your argument, but it can be made explicit for clarity.
- Section 4: (More formal "Limited data vs generalization"): I am still not sure how this is different from iid generalization, or how the lemma 4.1 makes this point. My understanding of lemma 4.1 is that there exists a context distribution for which the distribution of states induced by a collection of policies will never be equal to the BC policy. But this is because the policies collecting the data are conditioned on information not available to the BC policy. This induces something like partial observability, which is the source of the problem. Without the context, I do not think this would hold and thus stitching generaliation would be equivalent to iid generalzation. But the problem is, in a sense, limited data: the context is never accessible and hence the data (or information) is fundamentally limited.
- Section 4 (stitching is not finding an optimal policy): I agree that stitching can help find an optimal policy, but this is besides the point in the context of your setting (offline RL). The question is two-fold, under what conditions: 1) can stitching help find a better policy and 2) does fidning a better policy necessitate stitching.
- Section 5 (Nearby states) : how important is it that the states are mapped exactly onto the states in the set of experience? For example, you could imagine adding noise to all states and goals, which would therefore naturally stitch several nearby trajectories. Given the fact that you are assuming the space to have a distance metric, it is probably well-behaved so that this is effective.
- Section 5 (conditions for evaluating stitching): While the first condition is easy to engineer into the problem, the second condition seems problematic. How can you know whether the BC policy has a non-zero probability besides running the experiment?
- Section 6 (DT + aug?): One interesting approach that is missing is combining DT with data augmentation, seeing as DT is more performant than RvS. Is there any reason why this was left out? Is it computational concerns or a more fundamental limitation.
- Section 6 (More data): This seems at odds with condition 1 under "popular datasets do not evalute stitching". Surely, if every state and goal pair were included in the dataset, then DT would improve?

---

> ### Author Response · Authors · 2023-11-16
> **Response to dPJE (1 / 2)**
>
> Dear reviewer,
> We thank the reviewer for their detailed feedback. It seems like the reviewer's main concerns are about the correctness of a theoretical claim (stitching vs generalization) and the limited experiments. We believe the theoretical results are correct (see below), and have revised Section 4, 5, and added a new proof in Appendix D.1 to clarify the results; we have also added new image-based experiments to bolster the empirical contributions of the paper. Below, we will describe these experiments in more detail and answer the specific reviewer questions **Do these answers, revisions and experiments address all the reviewer's concerns?**
>
> > Why is stitching a problem of generalizing to a different test distribution, and not a problem that occurs due to limited data from the training distribution?
>
> Consider two datasets collected via two different sources:
>
> Source 1 : Different data collecting policies $\beta_h$, with probability $p(h)$.
>
> Source 2 : The BC policy $\beta$.
>
> Both these datasets induce the same discounted distribution over states (Lemma 3.1), but the induced distribution over trajectories (state - goal pairs) is different in general, even after collecting infinite data from both sources (Lemma 4.1). This can be observed from Figure 1, where there are 4 possible trajectories. The BC policy $\beta$ induces a distribution that samples all 4 trajectories (Figure 1.c),. But the data collecting policies $\beta_h$ induce a distribution over only 2 trajectories (Figure 1.b). This holds true even if  infinite data is collected from both sources. We agree that if we had access to the other 2 trajectories, then OCBC algorithms would be able to perform stitching generalization. Our claim is that these trajectories are out of training distribution, because we assume that the data is collected using Source 1. If we have access to all possible out-of-distribution trajectories, then of course generalization wouldn’t be an issue. Moreover, the perspective that, “if only you had data from the test distribution the problem would be solved, and hence it is a limited data problem” is incorrect when the train and test distribution are different. Because you will never have data from the test distribution by definition. And since the test distribution for stitching is different from the train distribution (Lemma 4.1), stitching is a generalization problem.
>
> > I think there are a few details that seem inconsistent throughout the paper, such as the fact that data is not the fundamental limitation vs popular datasets not evaluating stitching because they include most (s,g) pairs.
>
> We want to clarify that this is not an inconsistency.  In section 5, in para “Popular offline datasets do not evaluate stitching” we have added one sentence (in green color) to clarify this. The main claim of our paper is that:  In offline RL settings, when you have multiple data-collecting policies, even if you collect infinite data from these policies, generally there will be state-goal pairs that would never be sampled together. At test time, we need to evaluate the algorithm on exactly these state-goal pairs to evaluate stitching. The problem with the original D4RL datasets is that they evaluate on the same state-goal pairs, which are visited by the same data collection policy. See Figure 10 (D4RL) vs Figure 3 (ours) to see this difference.
>
> > experimental evaluation seems limited overall, and somewhat preliminary.
>
> We have added an image-based version of the point maze environment which uses a top-down image of the maze instead of the low-level x,y coordinates (see Figure 9). Figure 9 reaffirms that OCBC algorithms are unable to stitch. And despite the high dimensionality, our data augmentation does perform stitching. We would like to add that our experiments on Ant locomotion tasks, which are very popular in offline RL literature [1,2,3], and are considered to be challenging (much harder than the D4RL’s locomotion tasks).
>
> > Stitching as goal-state pairs : My understanding of the stitching property is that it is more general
>
> We agree that the term stitching is used more generally. At the start of section 4, we have added a paragraph (green text) that talks about different properties which are associated with stitching. Our definition indeed covers all the existing trajectories that can be re-combined using dynamic programming. In the offline RL setting, dynamic programming methods compare all actions (be it from different trajectories) that were taken in a particular state. By definition, these actions come from the distribution of the BC policy. This is exactly the distribution used to test for stitching. Our definition can be trivially extended to the reward-case, by substituting state-returns pairs (similar to the Decision Transformer paper) instead of the state-goal pairs.

---

> > ### Author Response · Authors · 2023-11-16
> > **Response to dPJE (2 / 2)**
> >
> > > In a way this is a source of limited data, if the "data" is a trajectory rather than an experience tuple. I think this is implicit in your argument, but it can be made explicit for clarity.
> >
> > We would like to reaffirm that this is not a source of limited data. For example in Figure 1, even after collecting infinite data from the blue and red data collecting policies, we would never see (state 2, goal 4) together. The reason why this isn’t a problem of partial observability is that you do not need the “context” information to recover the distribution of the BC policy. TD learning methods also do not have access to the context information, but can still perform stitching.
> >
> > > I am still not sure how this is different from iid generalization,
> >
> > Stitching is defined as a different form of generalization (not iid) because test and train distribution for stitching are different. You are correct to say that if the offline data was collected by a single data collecting policy then the train and test distribution would become the same. The case where offline data is collected by a single data collecting policy,  is covered by our framework where there is just one context with probability 1. For example, in Figure 1, if the underlying dataset was actually collected by the purple policy, we would eventually see (state 2, goal 4) together.
> >
> > > how important is it that the states are mapped exactly onto the states in the set of experience?
> >
> > It is not important to have exactly similar states. In our ant experiments, the observations are R^29 and it is very unlikely that the same floating point number is repeated twice. Adding noise to states and goals  will not make the test and train distributions over states, goals similar. In Figure 3 (left), in the pointmaze umaze dataset, a goal in the yellow region will never be sampled from a state in the red region. Also, adding larger amounts of noise will sample invalid states (for eg walls). Additionally, our new image experiments (Fig 9 and Fig 10) suggest that our method also works well in high dimensional settings. We welcome your suggestions and are happy to conduct an experiment if you can give us a way to add noise.
> >
> > >  The question is two-fold, under what conditions: 1) can stitching help find a better policy and 2) does fidning a better policy necessitate stitching.
> >
> > Intuitively, stitching enables OCBC algorithms to improve their performance on state-goal pairs which wouldn’t be seen otherwise. For example in Figure 1, without stitching, the OCBC policy would never be able to navigate from state 2 to state 4. To answer both your questions : for state-goal pairs that only occur across trajectories of different data collecting policies, stitching is both necessary and sufficient to find a better policy. In general, for any state-goal pairs, we can show that the optimal stitching policy is an improvement over the BC policy (See appendix D.1).
> >
> > > While the first condition is easy to engineer into the problem, the second condition seems problematic.
> >
> > While stitching is a good property for algorithms to have [1,2, 4], we don't always want to optimize for it in isolation for real world datasets. To evaluate whether your algorithm can stitch, it is easy to create datasets satisfying both conditions. For example, in Figure 3, test on state-goal pairs which lie in different colored regions. Moreover, we have open sourced our datasets, and code to create these new such datasets for future researchers (see supplementary material).
> >
> > > One interesting approach that is missing is combining DT with data augmentation, seeing as DT is more performant than RvS.
> >
> > Because DT [5] has memory, the data augmentation method should cluster a history of states and actions, which was difficult to do computationally. In the caption of Figure 4 (green text), we have added this clarification.
> >
> > > This seems at odds with condition 1 under "popular datasets do not evalute stitching". Surely, if every state and goal pair were included in the dataset, then DT would improve?
> >
> > As we explained above, if all state and goal pairs are already present in the dataset, then you won’t be able to test for stitching because everything will be in-distribution. The problem with the original D4RL datasets is that they evaluate on the state-goal pairs, which are visited by the same data collection policy. Hence it is possible for DT to perform well because it can capture the training distribution well. This is unlike our datasets, where there is a clear difference between methods that perform stitching (Data augmentation) and methods that don’t (OCBC).
> >
> > [1] OFFLINE REINFORCEMENT LEARNING WITH IMPLICIT Q-LEARNING
> >
> > [2] Optimal Goal-Reaching Reinforcement Learning via Quasi Metric Learning
> >
> > [3] HIQL: Offline Goal-Conditioned RL with Latent States as Actions
> >
> > [4] When does return-conditioned supervised learning work for offline reinforcement learning?
> >
> > [5] Decision Transformer: Reinforcement Learning via Sequence Modeling

---

> > > ### Author Response · Authors · 2023-11-20
> > > **Rebuttal Follow-up**
> > >
> > > Dear Reviewer,
> > >
> > > We hope that you've had a chance to read our responses and clarification. As the end of the discussion period is approaching, we would greatly appreciate it if you could confirm that our updates have addressed your concerns.

---

> ### Comment · Reviewer_dPJE · 2023-11-21
> **Thanks for the clarification**
>
> Both the changes to the paper and the answers to my questions clarify my misunderstandings. My new understanding is that stitching generalization is different from iid generalization because there is a latent structure in the (state,goal) pairs seen during training and those that are relevant for testing. Thus, the distribution is not identical nor independently distributed. The edge-case that I described, where all states and goals are present in the dataset, would not require stitiching generalization (and presumably, the supervised learning approaches would work here).
>
> I appreciate your reply. A few questions remain around the relevance of this problem setting, because the stitching property is not one that has been investigated before given the state of the very common offline RL benchmark (D4RL). These is value, however, in pointing out this property and I think this work need not answer that question. I will update my score (5 -> 6) to reflect this exchange.

---

> ### Author Response · Authors · 2023-11-21
> **Response to dPJE**
>
> We thank the reviewer for their comments and further discussion on the work.
>
> We believe the "relevance of this problem setting" is very important to the RL, supervised learning, and planning communities.
>
> > A few questions remain around the relevance of this problem setting, because the stitching property is not one that has been investigated before given the state of the very common offline RL benchmark (D4RL).
>
> The D4RL paper [1] has stated multiple times that testing for stitching is important (stitching is mentioned 6 times, See section 4,5 of [1] ). This raises an important question of when and whether stitching occurs. The first important contribution of our paper is to give a concrete definition of stitching as a form of generalization. Given this relation, we build on D4RL datasets to explicitly test for stitching. Indeed, our experiments reveal that OCBC algorithms which perform well on the original D4RL datasets [8], cannot perform stitching. We open source our code and the datasets needed to test for stitching. Arguably, stitching is the key property that helps offline RL algorithms find better solutions than the behavior policy. See appendix D.1, where we prove that the stitching is required to show one-step policy improvement over the behavior policy.
>
> Given the recent advent of using large supervised learning transformer models for many problems in RL, it is important to ground these approaches. This has become a very popular and relevant topic in RL [2,3,4,5,6,7,8,9] as well as in SL [10,11,12,13]. Our work shows both theoretically and empirically, why OCBC algorithms, even with larger transformers and larger datasets will not perform stitching. Our concrete definition of stitching, and open source datasets will give future researchers a chance to improve OCBC algorithms. While our work shows that current OCBC algorithms do not stitch,  we also propose a simple data augmentation which outperforms the baselines on all tasks on state-based (Fig 4) as well as image-based tasks (Figure 9,10). **In the light of these contributions, we believe that many RL researchers at ICLR 2024 would find our work useful and interesting. We kindly ask the reviewer to reconsider the paper in light of the revisions and clarifications.**
>
> [1] D4RL: Datasets for Deep Data-Driven Reinforcement Learning
>
> [2] Imitating Past Successes can be Very Suboptimal
>
> [3] When does return-conditioned supervised learning work for offline reinforcement learning?
>
> [4] You can’t count on luck: Why decision transformers and rvs fail in stochastic environments
>
> [5] Dichotomy of Control: Separating What You Can Control from What You Cannot
>
> [6] Upside-Down Reinforcement Learning Can Diverge in Stochastic Environments With Episodic Reset
>
> [7] Free from Bellman Completeness: Trajectory Stitching via Model-based Return-conditioned Supervised Learning
>
> [8] RvS: What is Essential for Offline RL via Supervised Learning?
>
> [9] Is Conditional Generative Modeling all you need for Decision-Making?
>
> [10] Faith and Fate: Limits of Transformers on Compositionality
>
> [11] Location attention for extrapolation to longer sequences
>
> [12] Unveiling transformers with LEGO: a synthetic reasoning task
>
> [13] NaturalProver: Grounded mathematical proof generation with language models.

---

### Author Response · Authors · 2023-11-16
**General response to all reviewers.**

We thank all reviewers for  your insightful feedback that has given us the chance to improve the paper. We would like to highlight some new updates and some key points that we feel are important to judge the paper:

1) The main contribution of our paper is to show that a desirable stitching property is fundamentally different from iid generalization. To the best of our knowledge, this has not been shown in prior work. We define this stitching property concretely as stitching generalization.

2) As a result of this relation, we show that OCBC methods do not have the stitching property. We verify this claim empirically (state based tasks Fig 4, image based tasks Fig 9 and Fig 10).

3) We propose a simple fix, a form of data augmentation. New experiments show that this works well on image-based tasks (Fig 9, 10).

The main takeaway of the data augmentation is that it should be used as a starting step to develop scalable and simple solutions that endow OCBC methods with these desirable properties that are often associated with TD-based methods.

---

### Comment · Area_Chair_CAKc · 2023-11-22
**Responding to Authors**

Dear Reviewers,

Please confirm that you have responded to the author's rebuttal. Thanks!

Best,
AC

---

### Meta-Review · Area_Chair_CAKc · 2023-12-10

**Metareview:**

This paper explores the link between this stitching ability and a specific type of generalization in goal-reaching scenarios, suggesting that data augmentation can enable these algorithms to effectively 'stitch' experiences and handle new challenges.

The reviews are split. One negative reviewer raised the initial score after the rebuttal period. Other negative reviewers are concerned about whether the required/assumed metric exists. Although there is a major drawback in the current paper (and should be made clear in the first paragraph of the introduction or the abstract to avoid misperception), I find the theoretical insight (although has a major drawback) useful, as the “RL as SL” literature remains less explored theoretically. The empirical validation appears informative to support the theoretical insight.

**Justification For Why Not Higher Score:**

See the above meta-review for the major drawback.

**Justification For Why Not Lower Score:**

Although there is a major drawback in the current paper (and should be made clear in the first paragraph of the introduction or the abstract to avoid misperception), I find the theoretical insight (although has a major drawback) useful, as the “RL as SL” literature remains less explored theoretically. The empirical validation appears informative to support the theoretical insight.

---

### Decision · Program_Chairs · 2024-01-16

Accept (poster)